# Robotic-Based Well-Being Monitoring and Coaching System for the Elderly in Their Daily Activities

**DOI:** 10.3390/s21206865

**Published:** 2021-10-16

**Authors:** Francisco M. Calatrava-Nicolás, Eduardo Gutiérrez-Maestro, Daniel Bautista-Salinas, Francisco J. Ortiz, Joaquín Roca González, José Alfonso Vera-Repullo, Manuel Jiménez-Buendía, Inmaculada Méndez, Cecilia Ruiz-Esteban, Oscar Martínez Mozos

**Affiliations:** 1ETSII (Escuela Técnica Superior de Ingeniería Industrial), Technical University of Cartagena, St. Dr. Fleming, s/n, 30203 Cartagena, Spain; francisco.ortiz@upct.es (F.J.O.); jroca.gonzalez@upct.es (J.R.G.); jose.vera@upct.es (J.A.V.-R.); manuel.jimenez@upct.es (M.J.-B.); 2AASS (Applied Autonomous Sensor Systems), Örebro University, 70281 Örebro, Sweden; eduardo.gutierrez-maestro@oru.se (E.G.-M.); oscar.mozos@oru.se (O.M.M.); 3The Hamlyn Centre for Robotic Surgery, Imperial College London, London SW7 2AZ, UK; dbautista432@gmail.com; 4Department of Evolutionary and Educational Psychology, Faculty of Psychology, Campus Regional Excellence Mare Nostrum, University of Murcia, 30100 Murcia, Spain; inmamendez@um.es (I.M.); cruiz@um.es (C.R.-E.)

**Keywords:** assistive robotics, affective computing, ambient assisted living, smart home, mood prediction, mental well-being, quality of life, ecological momentary assessment (EMA), machine learning, ROS

## Abstract

The increasingly ageing population and the tendency to live alone have led science and engineering researchers to search for health care solutions. In the COVID 19 pandemic, the elderly have been seriously affected in addition to suffering from isolation and its associated and psychological consequences. This paper provides an overview of the RobWell (Robotic-based Well-Being Monitoring and Coaching System for the Elderly in their Daily Activities) system. It is a system focused on the field of artificial intelligence for mood prediction and coaching. This paper presents a general overview of the initially proposed system as well as the preliminary results related to the home automation subsystem, autonomous robot navigation and mood estimation through machine learning prior to the final system integration, which will be discussed in future works. The main goal is to improve their mental well-being during their daily household activities. The system is composed of ambient intelligence with intelligent sensors, actuators and a robotic platform that interacts with the user. A test smart home system was set up in which the sensors, actuators and robotic platform were integrated and tested. For artificial intelligence applied to mood prediction, we used machine learning to classify several physiological signals into different moods. In robotics, it was concluded that the ROS autonomous navigation stack and its autodocking algorithm were not reliable enough for this task, while the robot’s autonomy was sufficient. Semantic navigation, artificial intelligence and computer vision alternatives are being sought.

## 1. Introduction

Scientific advances, in general terms, have improved living conditions as reflected in such basic aspects as longevity. According to several studies, there are expected to be around 125 million people over the age of 65 in the EU in 2030. Life expectancy has indeed improved considerably in recent years. Spain is one of the countries at the top of this list despite a decline of 1.6 years in 2020 due to the COVID 19 pandemic, as has happened in other countries [1]. 

Increased longevity is not always accompanied by greater happiness. [2]. Cities are getting bigger and bigger and at the same time there are more lonely people. According to the survey published by the INE (the Spanish National Institute of Statistics) 42.7% of women over the age of 85 lived alone, compared to 23.6% of men [3]. Since humans are naturally social, this is becoming a significant psychological problem [4,5,6]. A clear example is the current COVID-19 pandemic, which has brought about social isolation with repercussions on mental health. Lockdowns and quarantines have forced even more of the elderly to live alone. INE statistics show that the number of people over 65 who lived alone in 2020 was 2,131,400 compared to 2,009,100 in 2019 [7], so that loneliness is already postulated as one of the main epidemics of the 21st century.

Despite the loneliness involved, older people defend their right to stay at home and in many cases they refuse to move in with relatives or to go to residences for the elderly, even though their mobility and ability to look after themselves could be progressively reduced [8]. Although assistance can be given by home caregivers, it will be difficult to cover all demands for home assistance in the near future due to a shortage of available health workers and doctors as a result of greater life expectancy and low birth rate. There is thus an urgent need for innovative forms of support and health care for the elderly to maintain their physical and mental well-being, for which society should be prepared by adapting to the greater health assistance and monitoring requirements.

The latter is an advantage since it allowed reaching populations with scarce resources, difficult to reach by traditional means [9]. In recent years, the contributions of psychology have made sense with the intention of promoting interest in health and well-being during aging. Being relevant that people live longer, with quality and above all with autonomy, health and well-being [10].

The RobWell Project was begun in 2019 with the aim of finding technological solutions to these needs. The team is made up of a multidisciplinary group of researchers, engineers and psychologists from the Technical University of Cartagena, the University of Murcia (Spain) and the University of Örebro (Sweden) and the University of Kuyshu (Japan). It is part of the larger HIMTAE Project in cooperation with the University Carlos III of Madrid, which includes physical assistance in the kitchen with a manipulating robot (not described in this paper). The RobWell Project includes a mobile robotic platform integrated with ambient intelligence in a smart home that also includes estimating the user’s mood through wearables with a proposal for emotional coaching strategies.

From the user’s point of view, the idea is to create a relatively simple system that can be installed in homes to monitor the user’s daily activities and his/her state of health and mood by means of distributed home automation sensors, medical devices and smart bands. The data is interpreted by artificial intelligence to estimate the habits and state of mood of those living alone. If strange behavior or a low mood is detected, emotional coaching strategies are proposed by a small robot with the size of a robot vacuum cleaner [9] through smart speakers. Alerts can also be generated to reach caregivers, family members or emergency services if necessary. Strange behavior would be considered, for example, as spending too long in bed, not going to the bathroom for a long time, or not opening the refrigerator all day. Emotional coaching strategies are recommended such as: “You should go for a walk”, “Why don’t you call someone on the phone” or “You haven’t eaten anything for a long time”. The robotic-based ambient intelligent system will not only detect situations that suggest that the person needs help but can suggest activities to improve the person’s mood, such as leaving the house to meet other people.

This system requires expertise ambient assisted living, sensor data analysis, and machine learning. There is an additional distributed artificial intelligent system that coordinates all the subsystems, with symbiosis of physical and virtual robotic and AI subsystems to assist the elderly living alone during their daily activities. Advances in smart home devices and activity wristbands may make it possible to address the design of assistive environments with low-cost devices to achieve a relatively simple system that can be marketed at an affordable price to potential users.

One of the most important aspects of this project is the artificial intelligence system involved in mood recognition. As we pointed out in a previous paper [10], good results have been obtained in laboratory studies [11,12]. However, there are many problems with making these experiments a reality. The biggest impediments encountered are the number of uncontrollable variables involved in daily activities and the inconvenience of including the sensors used in the studies in daily tasks. Advances have been made in wearable devices, including the possibility of transferring these studies to other scenarios (such as clinics [13] or experiments without laboratory models [14]). Ecological momentary assessments (EMAs) have been used in this project as proposed in [10].

The main objective of this article is to present the RobWell project, including the system architecture and integration of the distributed sensor ecosystem, data collection and method of mood prediction using machine learning algorithms. To do so, the following points will be discussed:General scheme, both software and hardware proposed. This section will present the general hardware/software composition including elements such as the robotic platform, the sensors for home automation and user monitoring, the actuators, the data acquisition system for mood estimation, the user interface and so forth.Integration of the proposed elements. The proposed test system and the integration achieved in the field of mobile robotic platform and home automation will be discussed.Robustness of the navigation of the robotic platform. The navigation system implemented on the robotic platform will be discussed together with the continuous operation tests. In this way it will be possible to discuss whether autonomous navigation needs more sophisticated navigation strategies. The autodocking algorithm offered for the Kobuki robotic base will also be tested for robustness in continuous operation. With the continuous running test, the autonomy of the robotic system will be observed.Mood prediction. The data acquisition system designed for conducting the experiment in an everyday environment is discussed. On the other hand, the machine learning strategy to be used for mood prediction is also discussed. In addition, the psychological tools to be used (questionnaires and EMAs) are presented.

## 2. Related Works

The different technologies involved in the RobWell Project coexist and interact with each other to achieve the ultimate goal: the user’s well-being. Four major groups address the current state of the art: Assistive Robotics, smart home and Ambient Assisted Living and Mood Recognition by means of biomedical signal processing.

### 2.1. Assistive Robotics

This field is attracting growing interest and is now focused on care in hospitals, nursing homes, rehabilitation clinics and even people’s homes. In this way, the sick, disabled or elderly can have continuous care and supervision. Assistive robotics can include ADL (Activities of Daily Living) tasks such as cleaning, administering medication and assisting healthcare personnel, general assistance and feeding, among others. Some outstanding ADL projects include the Hospi Rimo robot [15], already installed in many Japanese hospitals, or Moxi [16], which is being tested in hospitals in Texas.

Tasks such as feeding and personal hygiene represent a great challenge in terms of assistive robotics. In the case of feeding, there are challenges such as the manipulation of the food itself and tools of different geometry and size. One of the latest advances in assisted feeding is from the University of Washington [17]. With regard to personal grooming, there are designs for robotic toilets with a wall mounted motorized chair with three degrees of freedom [18] while the Cody robot provides a solution to keep bedridden patients clean [19].

In the field of mobile robotics applied to monitoring the elderly, Samsung proposes Bot Care [20], a small robot that aims to ensure that the owner is in an optimal state of health by analyzing data such as breathing, heart rate, activity levels, stress, quantity and quality of sleep, and so forth, alerting emergency services if necessary. Researchers at the Technical University of Valencia, Spain, have developed a similar low-cost robot that is able to recognize emotions, remember medication schedules and analyze information from biometric sensors such as wristbands that measure vital signs [21]. Another example is the PAL Robotics’ TIAGo robotic assistant [22]. Its functions include giving pills at the programmed time, monitoring physical condition and vital signs, and locating keys and mobile phones. It can also advise when it is time to eat, carry objects, recommend healthy dishes, assist the elderly person at bedtime, tuck them in or help them into bed.

### 2.2. Smart Home

The term smart home is defined in the literature according to the field of application to which it refers. Attempting to group and generalize the definition, Ehsan Kamel and Ali M. Memari [23] determined that a smart home is a dwelling in which data related to the home environment and its residents are obtained from sensors, electrical devices or home gateways, and transferred using communication tools and networks for the purpose of monitoring devices and executing units, either to help decide actions or for the execution of so called services, these being the activities carried out by the smart home. Although there is no unified and definitive criterion for the differentiation of a smart home based on services, many authors agree on the following fields: health, well-being, security (internal agents), safety (external agents), entertainment and energy management [24]. In most cases, the name of the service speaks for itself. However, a distinction can be made between wellness and health, with the former referring to trying to act before an illness occurs and the latter to monitoring the user’s vital signs to provide data to medical staff, promoting telemedicine. With regard to safety, a clarification is made in brackets. Internal agents refer to those found inside the home (leaving the iron plugged in or the cooker on) While external agents detect intruders. 

The project described in this paper focuses on health and well-being, even though it has additional functions. As already mentioned, it is oriented towards the care of the elderly. Similar projects found in the literature include the following: in 2015, Tokyo Denki University proposed a home health monitoring system for elderly people living alone. This system focuses on monitoring three daily activities to differentiate between urination, kitchen activities and personal grooming activities. Sensors such as infrared for movement and flow sensors for the other activities are used for this purpose [25]. In 2017, the SMARTA project was launched by the LAM (Movement Analysis Laboratory). The main objective of this project was to develop and test a personal health monitoring system that integrates standard sensors, innovative wearable and environmental sensors. By studying these data, anomalies in everyday activities could be detected. Some of the vital parameters monitored included ECG, blood pressure, oxygen saturation and ear temperature, among others. In terms of environmental monitoring, they focused on the use and status monitoring taps, dishwashers and refrigerators [20,26]. In 2016, a doctoral thesis was defended at the University of Seville on integrating a service robot in a smart home using the UPnP (Universal Plug and Play) protocol [27]. Another similar project proposed the integration of the CSRA (Cognitive Service Robotics Apartment) mobile robot with an anthropomorphic top into a smart home. In this project, the aim was to achieve a human-machine interface as homogeneous as possible for access to sensors, actuators and services. For this purpose, the RSB (Robotic Service Bus) middleware was proposed, a new event based, message oriented middleware in a logically unified bus with a hierarchical structure [28,29]. In 2016, SmartSEAL was also proposed, an interconnection system for the SEAL research project, which focuses on developing solutions for home automation in the field of energy control, user adaptability and security enhancement [30]. This project starts by assuming that there are a large number of commercial communication standards for each device and that, in many cases, the integration between these devices is complex (interoperability and integration problems in heterogeneous systems). Although the use of assistive robots for integration is not mentioned in this case, what is really interesting is the use of ROS as middleware for the integration. For this, it proposes an architecture based on three types of nodes to differentiate between translator nodes to translate between protocols, cooperative nodes to pool elements from different manufacturers and the database manager node. Further information on this protocol can be found in [30].

To conclude the review of related articles, it is worth mentioning Reference [31], which presents an ROS based Integration of Smart Space and a Mobile Robot as the Internet of Robotic Things involving the new concept of IoRT (Internet of Robotic Things). This term encompasses and combines the terms robotic cloud and IoT. It shows how the key features of robotic technology, manipulation, intelligence and autonomy are related to scenarios in which IoT is applied. To do this, compatible communication protocols are used to connect the robot (or robots) with the smart space [32]. This project aims to demonstrate how a smart home can provide motion assistance through robots that do not have vision systems, image processing and intensive calculations for decision making, to prove that sensors mounted on the robot can be replaced by others found in the environment [31]. One of the most interesting projects that has used ROS as middleware dates back to 2017 [33]. In this project, a general architecture was designed and implemented for the integration of different elements with reasoning capable planning modules for application to a smart office. This project is important because it implements a similar idea to the one proposed in our project, including the same mobile robot (Turtlebot). It is important to mention that this system has been tested in an environment emulating a smart office. In the case of the HIMTAE Project, the environment is an elderly person’s house, which can become more complex. However, future work in this article reflects a number of practices to investigate that may be interesting to carry out our system, such as having different coordinated ROS masters or attempting mutual cooperation of different elements (an event addressed in the HIMTAE/RobWell Project [33].

### 2.3. Wearables in Affective Computing

Emotional assessment and regulation has been one of the promising wearable applications since the pioneering works of Picard and Healey, and the presentation of their affective sensing device at the first IEEE International Symposium on Wearable Computers in October 1997 [34]. According to these authors, wearables allow continuous emotional state estimation as they can be in close, long term physical contact with the user, enabling measurement of their condition on a scale of observations per person per minute rather than in a period of months or even years through current metrics and methods. As Pentland later pointed out in Social Physics [35]. This specific characteristic could also enable long term data gathering for an individual “in the wild”, rather than the usual short-term data gathering for a group of people in the laboratory.

After these initial basic technology research works (Technology readiness level -TRL- 1), it was just a matter of time for the technological offer to reach the maturity level re-quired for the proposal of research projects for evaluating the feasibility of using these systems as feedback sources for human-robot interaction (TRL 2–3). In this regard, the Universal Emotion Recognizer (UER) used in the psychophysiological control architecture for human-robot coordination proposed by Sarkar in 2002 [36] can be considered as the first integrated device for personal emotional assessment after demonstrating its potential for inferring the emotional state of a particular individual after her observed physical expressions in a given perceivable context. Later works by this team at Vanderbilt University proposed the development of personal robots designed to act as understanding companions to humans [37], boosting research in the application of psychophysiology measurements for Human-Robot Interaction [38].

When devices such as the first FitBit were introduced, authors such as Swan clearly identified the potential of these wearables for developing new patient driven health care services, including those related to patient quantified self-tracking for health applications and emotional support [39]. 

With the advent of Ambient Intelligence (AmI) [40] and Ambient Assisted Living -AAL- [41], initial requirements for user-centered design of Ambient Assisted Emotional Regulation systems [42] were proposed after the usability analysis of this first generation of commercially available smart wearable devices [43]. 

The electronics industry quickly reacted to the consumers’ expectations, increasing the availability and diversity of new electronic sensors specially conceived for integration in the second generation of these devices [44]. Besides the “classical” MeMs-based accelerometers, inertial units (integrating triaxial magnetometers, gyroscopes and accelerometers), pulse sensors and biopotential front-ends developed by the main industry stakeholders, enabling the design and commercialization of advanced experimental wearable devices [45] intended for sampling biosignals at higher frequencies with better resolution [46]. 

The adoption of open operating systems, APIs and SDKs contributed to the spread of these technologies, boosting the development of new systems and solutions [47], that later lead to the advent of the era of the Internet of Wearable Things (IoWT) [48]. The design of this new generation of devices (TRL 9) considered not only functional requisites but also others such as aesthetics and usability [49] to extend their use to other population groups such as the elderly [50,51,52] raisings user awareness on other emerging topics such as privacy related to lifelogging [53]. 

As happened previously with the spread of smartphone use, after the introduction of these advanced devices, research was then oriented toward the development of health care solutions [46,54,55]. Initially, most of them were centered on clinical assessment of chronic conditions [56] and neurodegenerative diseases such as Parkinson’s disease [45,57,58,59] or Multiple Sclerosis [60,61,62,63,64,65,66]. These studies contributed to the knowledge related to subrogated alterations in the user’s condition (e.g., balance, gait or sleep patterns, etc.) useful for diagnostic and screening purposes. Besides these clinical results, these research activities have made other technical and non-technical factors visible, affecting the final outcomes of these projects.

Among the technical factors directly related to the success of these studies, device power consumption, battery duration and recharge time have arisen as critical topics to be resolved by manufacturers to reduce system down time, since battery charging disables data acquisition in most devices [44,67,68]. Other factors affecting the quality of the experience of smart wearables [69] include material wear, battery deterioration, interface miscommunication, data inaccuracy, tracking limitations, software glitches and dysconnectivity [70].

User engagement and motivation have also been found to be critical [71], particularly in the absence of meaningful feedback [72] as happens in blind studies [73,74,75].

#### 2.3.1. Wearables and Emotional Biomarkers

Most of emotion or affective estimation projects are based upon the analysis of subrogated physiological responses related to the different dimensions of the emotional model proposed by Russell: valence and arousal [76].

When looking for emotional biomarkers, typical in lab experiments have involved recording psychophysiological responses of experimental subjects after the presentation of certain stimuli, either relevant or not. Tonic responses are measured as baseline values or resting levels of each signal in the absence of relevant stimuli while phasic responses gather the effect of certain stimuli in relation to the baseline values (i.e., increase/decrease, slope, etc.). In opposition to these, spontaneous or non-specific responses are measured when there is no known stimulus presented [38], being the typical kind of data acquired “in the wild”, as in this project.

Since the initials works on affective computing by Picard and Healey, researchers have tried to acquire the same biosignals set as that used in lab conditions, including respiration (chest expansion measured with resistive/hall sensors), skin conductivity (GSR galvanic skin response), temperature, photoplethysmography (BVP, blood volume pressure), heart rate (measured after the BVP signal/ECG) [34,36,42,77,78] and muscular activity (though the acquisition of the electromyogram, EMG,) [34,36], brain activity (through the electroencephalograph, EEG) [36,79] gait analysis [80] and others (inertial parameters, light level, etc.) [81,82,83]. An extensive review can be found in Saganowski et al. 2020 [84]. These sensors are intended to record the physiological biomarkers related to the level of arousal at both autonomic and central levels. 

Heart Rate. Heart rate is controlled by the autonomic nervous system, which comprises both sympathetic and parasympathetic (vagal) branches, whose action is normally balanced. When exposed to a stressor, this balance is lost, increasing the activity of the sympathetic branch, which leads to higher heart rates. In order to have a continuous indicator of this regulation, heart rate variability analysis is performed by studying the frequency contents of the spectral representation of the Inter-Beat-Interval time series (IBI) [85]. Mean and median frequency values of the power spectral density (PSD) of the IBI spectrum have been proposed as indicators [36]. Special care should be taken when performing the spectral transformation since this series is not evenly sampled by nature. Time-domain parameters such as RMSSD, NN50 and pNN50 are also considered [42,77,78,86,87].Respiration. As happens with heart rate, the respiration rate is also controlled by the autonomic nervous system, and thus is susceptible to be used as a subrogated indicator of the onset of stress and anxiety [34,57,85].Photoplethysmography. Easily acquired by means of non-invasive sensors, blood volume pressure (BVP) has been widely used in affective detection experiments and instantaneous heart rate calculation [34,42,57].Electromyography. Anxiety and stress can lead to increased muscled tension, as in the masseter [34] and the cervical trapezius [36], being muscle electrical activity (usually quantified as the root-mean value of the EMG signal) a subrogated measurement of these anxiety and stress levels [57].Skin Temperature. As above, finger blood capillaries vasoconstrict because of anxiety and stress, reducing the effective blood flow, which leads to decreased skin temperature, which is simple to measure using small mass thermistors [36,77].Galvanic Skin Response. Stress and anxiety lead to an increase in sweating activity, which reduces the electrical resistance of the skin, which can be determined by injecting a minimal known DC-current and then measuring the voltage across the electrodes [34,36,42,77].Gait analysis. Although gait analysis has been widely used for studying neurological degeneration related to Parkinson’s disease, Multiple Sclerosis and Alzheimer’s, only a few works have tried to find correlations of gait indicators with the emotional state of the subjects studied [80,88].Brain activity. Through there are some works related to emotion detention in-the-wild based on electroencephalogram (EEG) analysis, there are some serious concerns limiting the practical application of these methods. Most of them are related to the reduced signal-to-noise-ratio achieved in real life conditions due to the small amplitude of the recorded signals and their susceptibility to artifacts (i.e., eyeblinks, heartbeats, jaw and forehead muscle tension) and electromagnetic perturbations [79].Other. With the advent of smartphone integrated sensors, some studies have taken advantage of them to build huge lifelogging datasets including signals from the built-in inertial sensors (accelerometer, gyroscope ad magnetometers), compass sensors, environmental parameters sensors such as sound and light levels and air pressure, humidity, and temperature as well as other as location and phone state [81,82,83].

#### 2.3.2. Recommended Devices 

As mentioned before, most of these recent studies have used commercial devices, the best suited for these purposes being the relatively old E4 from Empatica and the no-longer available Microsoft Band 2 [84]. 

Despite this, it is probable that new devices, including these as well as other advanced sensors, will be made commercially available in the short term due to market trends. According to the latest forecast from Gartner, worldwide end-user spending on wearable devices will total $81.5 billion in 2021 (an 18.1% increase from $69 billion in 2020). It is interesting that other form factors are being considered for these devices such as ear-worn devices (forecast to reach $39.2 billion in 2021) and smart patches (forecast to reach almost $6 billion in 2021). This latter category, first introduced in this last report, is expected to include sensors enabling long term non-invasive monitoring capabilities as advanced by devices such as BioStampRC [62] and the more recent BioStamp nPoint.

Society has experienced several changes in the way technology is understood nowadays. The community of Artificial Intelligence (AI) is responsible for this transition to-wards a modern society [89]. Many works integrate advances in mood recognition using Machine Learning algorithms in laboratory settings [11,12]. One main drawback of the before-mentioned works is the procedure used to gather the data, which is counterproductive to the goal of this project: monitoring well-being during daily activities. Normally, the sensors used hinder daily activities. There is also another problem to tackle in this work: emotion monitoring. Traditional methods rely on retrospective reports which normally are subject to recall bias. As pointed out by Shiffman et al. [90] the aforementioned approach limits the ability to precisely characterize, understand and change behavior in real world conditions. One solution for this is the so-called Ecological Momentary Assessment (EMA), which eases the patients’ mental burden, since traditional tests for the assessment are too long to be asked repeatedly during the day. In addition to this, the growing capabilities of smartphones and wearable devices makes the use of the EMA approach much more feasible as a solution for monitoring mental illness, treatment, self-management, and intervention applications, thus expanding the coverage of mental health services by deploying economically friendly solutions.

The above-described method of gathering information on mental well-being has been used in several works. In [91], a system is proposed to monitor potential depressive patterns in the elderly living alone. Emotional state is collected from diverse sources such as: surveys, smart watches and EMA questionnaires. As mentioned above, the power of EMA methods to obtain emotional information can expand the boundaries of this research area to daily life applications. For example, in [92] a system which gathers information in the context of person’s activities can detect long-term stress patterns. Stress detection during daily real-world driving tasks through advances in intelligent systems is described in [14]. Techniques such as Machine Learning were also applied in [10] making it possible to monitor the elderly’s moods via intelligent sensors on a wristband.

### 2.4. Contributions

For the contributions of the proposed systems, the order of presentation of the related works will be followed. In the case of assistive robotics, a small robot that is capable of navigating autonomously around the house, avoiding both static and dynamic obstacles is proposed. The contribution comes from the ability to act on the user’s mood. The aim of this system (robotic platform) is to be able to navigate to the user to offer different options depending on the predicted mood. Some of the proposed systems [20] theoretically detect stress or have some form of music therapy implemented. However, this project proposes a robotic system that, based on the data collected through psychological questionnaires, is personalized to offer the most accurate coaching strategies. In addition to this, another contribution comes from integration. One of the main objectives is connectivity and cooperation between the systems. In this way, the aim is to achieve communication and interaction with the kitchen robot and for the home automation system to be able to feed off the information provided by the robots (and vice versa). Some of the related works present systems that include sensors from different manufacturers [30] and Turtlebot robot [33], but they do not talk about the cooperation of all these systems. Another important point to note in comparison to other systems is that low-cost commercial sensors are being used. In addition, being a system dedicated to elderly people, it was proposed to do it locally as cloud solutions were dangerous due to the loss of internet connectivity.

On the other hand, a method for acquiring data related to the user’s mood in everyday environment is also proposed. A data acquisition system based on a smartphone and the Empatica E4 medical device is proposed. In the case of the Empatica E4, it is a device that is unobtrusive to the conduct of everyday life. In addition, tests and questionnaires are answered through a developed Android application where users can organize their answering schedule. The ultimate idea is to get data from a few users but over a long period of time for each user.

## 3. Materials and Methods

### 3.1. Proposed System Design

#### 3.1.1. Hardware Architecture

The proposed system consists of different elements. As in any control system, a distinction can be made between sensors, control and actuators. In the hardware architecture of the proposed system, the different nodes (sensor, control and actuator) are composed of the elements shown in Table 1.

All these elements are represented in Figure 1. It should be noted that all these elements will be dealt with in depth in later sections.

#### 3.1.2. Software Design

The main problem with commercial devices is that, even though the devices may comply with the open Zigbee standard, each manufacturer has his own gateway and user application (Ikea, LIDL, Xiaomi, Philips, etc.). This makes it impossible to use devices with another brand’s gateways and the aim of this project was to integrate devices of different brands. One of the main software tasks was thus to provide inter-device interoperability and transparency.

As can be seen in Figure 2, the system is composed of a variety of software components for different domains. The elements that need specific software are as follows: robotic platforms, home automation system, positioning system, data acquisition system for predicting affective state and integration software. Of all these elements most are in the process of development or improvement, except for the indoor positioning system, which is currently under development. The rest of the systems are listed below with a brief description:Robotics platforms. Of the robotic platforms in Figure 1, only the emotional coaching platform will be explained because the other platform is being developed by the in the HIMTAE Project in the Carlos III University of Madrid. The teleoperated mapping, autonomous navigation, autodocking and power management software to determine its performance in continuous operation has already been implemented and tested. Further features and enhancements are under development.Smart home. The software that will run all the home automation logic is the Home Assistant operating system [93], selected because of its integration tools and the associated community for troubleshooting. It will run a Node-RED server and a Mosquitto MQTT broker. In addition, it also allows the use of the CC2531 USB dongle.Acquisition system for affective state prediction. As can be seen in Figure 1, at the hardware level, this system consists of an Empatica E4 medical device and a smartphone. Two Android applications are used for the extraction of the data used in mood prediction. The first one, E4 RealTime, is the official application offered by Empatica for the extraction of physiological data. There is also a self-developed application for conducting tests and questionnaires to relate physiological data to mood states. The final idea is that this whole process will take place in a single application.General integration system. This is the system in charge of carrying out the integration between the elements. There will only be differentiation at the level of sensors and actuators, regardless of the brand or manufacturer. This whole system will be developed on Ubuntu 16.04 and ROS. The use of ROS to carry out this task is justified by the fact that the integration, in some cases, is direct. Furthermore, being structured in nodes, the topics and following the publisher/subscriber policy makes development more accessible. The development of this application is not yet complete. For the time being, the integration paths have been basically developed and tested in a lightweight way. For example, the integration between Node-RED and ROS has been tested, as has the integration between elements with ROS, the integration between Zigbee and Node-RED elements with MQTT and between Android and Node-RED using MQTT. The integration application is currently in the conceptual development phase.

### 3.2. Home Automation

#### 3.2.1. Sensor Environment

In the introduction, we talked about the different types of smart homes associated with health and wellbeing. This leads to starting the selection of sensors according to the activities to be monitored. Elements belonging to other fields can also be added such as security (from both external and internal agents), entertainment or energy saving, all depending on the user’s needs.

The sensor environment consists of a set of commercial and self-developed sensors whose main function is to monitor the environment surrounding the user in the least intrusive way possible. When we talk about intrusiveness, in this case we refer to possible works or modifications to the home. One of the objectives to keep in mind is that most sensors should be small, unobtrusive and wireless. Obviously, this is not going to be achieved in all cases as there are sensors that will require mains power. This monitoring is carried on mental and physical health (smart some oriented towards wellbeing and health). At this point, one can think directly of the typical activity bracelets that provide information on physiological parameters. While it is true that these are a valuable source of information for the prediction of the user’s affective state, there are many other sources in the home environment; for example, the frequency of personal grooming, opening the windows, raising the blinds to let the sun in, the time spent in bed or on the couch, and so forth (Table 2).

With the information obtained from these sensors, it will be possible to extract the necessary information about the user’s well-being and health. This would cover the health and wellness smart home part. In addition to this, there will be additional elements for security like gas, fire and water leaks detectors. The use of smart light bulbs and smart plugs is also proposed to achieve greater home automation and improve the household energy performance. 

So far only a few commercial sensors have been tested in the system in addition to those offered by the Kobuki’s Iclebo base. It should be noted that the Iclebo base sensors are included in this case as they are interconnected sensors in the system although they will not be dealt with in depth. However, everything related to the robotics platform (sensors, actuators and software) will be dealt with in depth later on. Table 3 shows the sensors that we plan to include in the system. As some devices are not yet available, we have decided to perform a test setup with the commercial sensors presently available. The column headed “Inc.” specifies the sensors included in the test system (“Y”) and those which are not (“N”).

As can be seen in Figure 2, the communication protocols to be used are Zigbee, WiFi, MQTT and IR. Most of the commercial sensors have been selected with Zigbee protocol for power consumption reasons. If the system is to be as non-intrusive as possible, wireless devices should be selected. The Figure 3 shows the plan of the house where the system has been tested.

#### 3.2.2. Computer Platform for Home Automation

As can be seen in Figure 1 and Figure 2, the single-board low-priced Raspberry Pi 4 Model B is the computational node in charge of home automation. It was selected for its extensive use with Home Assistant and its good cost/performance ratio.

In this type of device, the operating system is usually mounted on a removable SD card. The problem is that in previous experiences with systems like those proposed in this article, it was seen that the memory card ended up being corrupted, so it was proposed to add an external SSD type memory. This requires an expansion board that allows the SSD memory to be integrated with the Raspberry Pi (Raspberry Pi Foundation, Cambridge, UK). For this case, the ONE M.2 case has been selected, which allows the integration of both elements (SSD and Raspberry Pi). The SSD memory selected was the Kingston A400 M.2 (Kingston Corporation, Fountain Valley, CA, USA).

As already mentioned, most of the sensors use the Zigbee protocol. In order to coordinate all elements of the system, a Zigbee Gateway is needed. Normally, each manufacturer offers a gateway for their branded products; however, the use of these gateways would make the system more closed and proprietary, not to mention that most of them offer solutions in the cloud and an internet connection is always needed. To overcome this problem, we propose the use of a generic CC2531 USB dongle (Texas Instruments, Dallas, TX, USA). It is a USB device on which a CC2531, a SoC for IEEE 802.15.4, ZigBee and, RF4CE related applications, are mounted. All the elements used can be seen in Figure 4.

The software used in this project is Home Assistant. This is a free, open-source software that is widely used for home automation. Home Assistant is characterized by the management of home automation locally. This provides greater security and speed compared to cloud-based systems. This software has a multitude of add-ons developed for the integration of different devices. It was therefore decided to install a Node-RED server, a Mosquitto MQTT broker and the necessary Zigbee2MQTT software for the management of the CC2531. Node RED is a programming tool that can be connected to hardware devices, APIs and online services through a programming based on flows and blocks. In this case it is used for integrating ROS. There is a node distribution in Node-RED called node-red-contrib-ros that allows us to publish and subscribe to ROS topics. There is also the Mosquitto MQTT broker. The MQTT protocol works according to the policy of the publisher/subscriber with the broker as the central block of the structure. Basically, it is the server that accepts the published messages and distributes them to the subscribing nodes. The Zigbee2MQTT add-on uses this broker to publish sensor information in MQTT topics. In this way, it will be easy to access this information from Node-RED and publish it in ROS. 

### 3.3. Robotic Platform

The Turtlebot II commercial solution has been chosen for the robotic platform. This is a mobile robotic base widely used in education and research. It is a differential type of mobile robot with two support wheels located in a rhomboidal position. The mobile base on which the robot is mounted is the Kobuki IClebo (YUJIN ROBOT Co., Ltd., Yeonsu-gu, Incheon, Korea) (Figure 5b) which can power most of the systems (computer, sensors and other devices). 

As this platform should map and navigate autonomously around the house, a Hokuyo UST-10LX LIDAR (Hokuyo Automatic Co., Ltd., Osaka, Japan) (Figure 5a) sensor was added to the base. In addition, the element that carries out robotics computing tasks will be a compact Intel NUC computer (Intel Corporation, Santa Clara, CA, USA) (Figure 5c), characterized mainly by its small dimensions. Another of its advantages is the possibility of being able to expand its features as required.

The robotic platform uses Ubuntu 16.04 as operating system and ROS as the software robotics framework. The mapping and navigation stack offered by ROS [94] (Figure 6) is used to map the home in which the robot is installed and autonomously navigate once mapped.

In addition to this, continuous operation experiments were carried out by monitoring the battery status. The package used for robot initialization is turtlebot_bringup [96] with a “.launch” file called minimal.launch, which launches the relevant nodes that publishes the desired information on the battery in a topic. Figure 7 shows the topic containing the battery information and field name. To interpret the evolution of the battery status, a node will be created that subscribes to the topic shown in Figure 7. Each time, the information of the desired data is updated, it will be saved in a text file together with its timestamp.

The idea of a continuous operation experiment consists of autonomous navigation to certain points in the house according to the time of day. The idea is to create time circuits that start and end at the charging station (Table 4). In this way, it is tested whether the autodocking algorithm provided by Kobuki is reliable enough to be applied for the proposed purpose.

### 3.4. Mood Estimation

#### 3.4.1. Acquisition and Processing of Physiological Signals

According to Figure 1 and Figure 2, this section proposes a system consisting of a device capable of sensing physiological variables and an application that proposes tests and questionnaires. Physiological and activity measurements are performed by means of an E4 wristband device from Empatica. This CE certified device, commercially available but intended for research, enables data acquisition of accelerometry, skin temperature, electrodermal activity and blood volume pulse signals for up to 32 h (in recording and data-logging mode). Streaming mode is used (24 h battery autonomy) in conjunction with a 4G internet connected smartphone, which is used as the gateway to the cloud storage services provided by Empatica. Information about the Empatica E4 sensors can be found in Table 5.

(a)Physiological Signal Processing and Feature Extraction

The different physiological signals were first preprocessed and filtered as summarized in Table 6. Features were extracted using a sliding window approach with window size of 60 s. This size has been widely reported as the appropriate value when working with physiological signals [14,97,98]. We also set an overlap of 10% between consecutive windows to reduce the boundary effect when signals are filtered.

(b)3-axis Accelerometer (72 features)

Three-axis accelerometer signals were sampled at a frequency of 32 Hz, and then the Euclidean norm of the acceleration vector was calculated before filtering the four time series with a 3rd order Butterworth band-pass filter with a 0.2 Hz to 10 Hz bandwidth [100]. Afterwards, features in the time-domain [85,100,101] and in the frequency-domain were extracted [102]. In Table 7, you can see the extracted characteristics of the accelerometer.

(c)Peripheral Skin Temperature (13 features)

The peripheral temperature signal is sampled at a frequency of 4 Hz. It is set a threshold of 2 °C to discard incorrect gathered data when detecting an increase or decrease on the skin temperature trespassing such threshold. Features in the time-domain [85,101,103] and frequency-domain [102] are then extracted. In Table 8, you can see the extracted characteristics of the skin temperature.

(d)Heart rate variability (27 features)

Since it was not possible to reliably reconstruct the tachogram after the IBI data read from the E4, an HR time series provided by the manufacturer was used for heart rate variability analysis. This instantaneous heart rate, sampled at 1Hz, was then normalized on a daily basis, taking the first ten minutes from each day as baseline to compare data from different participants. Features in both time-domain [85,100,101] and frequency-domain [104] were then extracted. In Table 9, you can see the extracted characteristics of the heart rate.

(e)Electrodermal Activity (82 features)

The EDA signal, sampled at 4Hz by the E4, was low-pass filtered through a 3rd order low-pass Butterworth filter at 1.5 Hz before extracting both, tonic (Skin Conductance Level, SCL) and phasic (Skin Conductance Response, SCR) components [99]. Features in both time [85,99] and frequency domains were calculated for each component [99,102]. In the Table 10 you can see the extracted characteristics of the electrodermal activity.

#### 3.4.2. Mobile App for Ground Truth

The application was developed for the Android operating system. This APP asks the user about two parameters: level of happiness and activity. To answer this question, the user must select the level of two bars divided into discrete levels, with 0 being not at all happy/active and 4 being very happy/active. The test interface can be seen in Figure 8a.

As the aim is to carry out five tests during the day, the application was designed so that the user could configure the times at which he/she wants to be notified to take the test. The notification alerts the user with an uncertainty interval of +− 10 min. The data extracted are as shown in Table 11.

All the information extracted through the application is saved in local files and Excel sheets in the cloud. Each user is assigned an ID code to identify him/her, so that no personal data is stored in the cloud. The designed application has two operation modes: user mode and administrator mode. The user mode only has the interface necessary to carry out the test, while the administrator mode allows the configuration of the test times (Figure 9).

As a significant amount of data must be obtained to train a machine-learning-based algorithm, labelled data are needed. In this paper, the EMA answer registered for each patient is extended a number of minutes before and after. We start out on the assumption that the preceding and subsequent physiological response stays in the body for this length of time before and after answering EMA using windows of 30-, 60- and 120-min.

As the participants can answer the questionnaires out of the time slot assigned to it, we can encounter the following situations: (1) the time between two consecutive answers is less than half of the fixed window; (2) the first and the last EMA of the day may be close to the time of the initial and final data collection. The procedure followed to solve these situations consists of evenly separating the available data and using it for each of the former situations, respectively. In Figure 10 we show an example of the explained situations.

#### 3.4.3. Signal Classification

The system introduced in this work is based on two elements: on the one hand, we have a wristband type device that can be used to obtain information on physiological variables and, on the other hand, we have an android application that collects user responses to the tests and questionnaires carried out [105]. This wristband is provided with sensors to monitor blood volume pulse (BVP), electrodermal activity (EDA), and peripheral skin temperature. It is also equipped with a 3-axis accelerometer and a built-in application to derive the heart rate (HR) and interbeat interval (IBI) from the BVP signal. 

The emotional state of the monitored person is collected following the model proposed by Russel et al. [76], which is a multi-dimensional approach that defines emotions by two dimensions: arousal (or activeness) and pleasure (or happiness), pleasure being the range of negative and positive emotions; and arousal representing their active or passive degree, as explained previously in Section 3.4.2. Therefore, each emotional state can be placed as a point; in this case, a bidimensional space (see Figure 8b). The collection of those two dimensions is done via a mobile app (see Figure 8a). The participant is asked to fill in the happiness and activeness felt at a certain time on a 5-point Likert scale to quantify his/her emotional well-being by asking only two concise questions, known as an Ecological Momentary Assessment (EMA). The small mental burden of this technique makes it suitable for the environment proposed in this work: emotion recognition during daily activities. The user is asked to answer this questionnaire five times a day and more answers can be included if required.

The extracted features from the signals and the emotional state collected with the mobile APP define the classification problem proposed to estimate mood states. This problem is following a supervised learning approach, being training data samples features vectors extracted by a 60-s window for each of the signals mentioned before; and labels the mood state collected by the APP as explained in Section 3.4.2. We refer to Figure 10 in the previous section to emphasize the integration of the mood state with the collected physiological data. Figure 10 is an example of the logic followed to assign every vector a label. The sliding window will be placed at every data point of which has a label value different to −1 (as reflected in Figure 10). Once every feature vector is assigned with a mood state, we proceed to train a Support Vector Machine (SVM) with the goal of mapping feature vectors into mood states by minimizing a loss function.

### 3.5. Psychological Instruments

In addition to the experiments explained in the previous sections to monitor emotional well-being, psychological questionnaires were also included. These questionnaires aimed at measuring the levels of stress, anxiety, depression, and general well-being at the beginning and end of the experiment. This is a positive aspect as the experimental setup (wearable sensor, APP to answer EMA) did not cause stress to the participants. In addition to the personal interviews at the beginning and end of the experiment, each participant answered the first 20 questions of the STAI questionnaire. This questionnaire provides potential insights into symptoms of depression. Frequency analysis was carried out to contrast the correlation between the arousal (or activeness) and pleasure (or happiness) dimensions with the anxiety levels registered. The results concluded that correlations between affective dimensions and anxiety state were not statistically significant. Detailed information of the tests during the data collection process is given below:A sociodemographic survey that provided information on age, gender, marital status and knowledge of digital devices (mobile, computer, etc.).A short Inventory of Emotional Intelligence for the Elderly (EQ-I-M20) by Pérez-Fuentes, Gázquez, Mercader and Molero [106]. This is an adaptation and validation into Spanish of the Bar-On and Parker [107] instrument for the elderly consisting of 20 items on Likert scale. It is made up of the following factors: (a) Intrapersonal; (b) Interpersonal; (c) Stress management; (d) Adaptability; and (e) General mood. It has a Cronbach’s alpha value of 0.89 for reliability on the general scale.CECAVIR (Quality of Life Assessment Questionnaire) prepared by Molero, Pérez-Fuentes, Gázquez and Mercader [108] consisting of 56 items ranging from 1 to 5 composed of six quality of life dimensions: Health, Social and family relationships, Activity and leisure, Environmental quality, Functional capacity and Satisfaction with life. It has a Cronbach’s Alpha of 0.865 on the global scale.STAI, State Trait Anxiety Questionnaire [109]. It is a questionnaire made up of 40 items with two scales. It can evaluate anxiety as a transitory state (Anxiety/state) and as a latent trait (Anxiety/trait). The alpha coefficient for State Anxiety ranges between 0.90 and 0.93 and for Trait Anxiety between 0.84 and 0.87 [109].Hamilton scale on depressive/anxiety symptoms (1960) [110] made up of 14 items on a Likert scale that ranges from 0 to 4 to evaluate somatic and respiratory symptoms, depression, and so forth. A higher score indicatives anxiety/depressive symptoms and provides information on psychic and somatic anxiety.Abbreviated Yesavage questionnaire (GDS) on depression in its Spanish version for the elderly [111], a questionnaire that evaluates depression in people over 65 years of age, made up of 15 items that score dichotomously (0–1). A total score over 5 indicates depression. It has an internal consistency of 0.994.Mini-cognitive exam -MEC35- [112], the Spanish adaptation of the Mini-Mental State Examination. It is an instrument that detects cognitive deterioration and examines different cognitive functions: orientation, memory, attention, calculation, language and construction, praxis and reasoning. The maximum score is 35 points, the optical cut-off points for cognitive impairment being in a population over 65 years of age and with a low educational level of 24 points.The Global Impairment Scale (GDS) by Reisberg, Ferris, de León and Crook (1982) [113], assesses the seven different phases of Alzheimer’s disease: stage 1 (normal), stage 2 (subjective memory complaint), stage 3 (mild cognitive impairment), stage 4 (mild dementia), stage 5 (moderate dementia), stage 6 (moderately severe dementia) and stage 7 (severe dementia). The internal consistency is very good, presenting a Cronbach’s alpha of 0.82 [113].The Katz index [114], which assesses the level of dependence/independence of daily activities. It has eight possible levels: (A) Independent in all its functions; (B) Independent in all functions except one; (C) Independent in all functions except in the bathroom or other; (D) Independent in all functions except in the bathroom, dress and any other; (E) Independent in all functions except in the bathroom, dress, use of the toilet and any other; (F) Independence in all functions except in the bathroom, dress, use of the toilet, mobility and any other of the remaining two; (G) Dependent in all functions; (H) Dependent in at least two functions, but not classifiable as C, D, E or F. Levels A–B (0–1 points) indicate absence of disability or mild disability, levels C–D (2–3 points) indicate moderate disability and E–G levels (4–6 points) indicate severe disability.Psychological well-being Scale (Díaz, Rodríguez-Carvajal, Blanco, Moreno-Jiménez, Gallardo and Valle, 2006) [115], an instrument that consists of 29 items that ranges from 1 (totally disagree) to 6 (totally agree). It has six subscales that evaluates self-acceptance, personal growth, purpose in life, positive relationships with others, environmental mastery and autonomy. It is Cronbach’s alpha ranges between 0.70 and 0.83 in all dimensions (Díaz et al., 2006) [115].Scales on positive and negative affect (PANAS), translated into Spanish by Robles and Páez [116] (2003), a scale composed of two factors of 10 items (ranging from 1 not at all or slightly to 5 a lot) that measures positive and negative affect and presents a Cronbach’s alpha of 0.92 for positive and 0.88 for negative affect (Robles and Páez, 2003) [116].NeO-FFI Questionnaire (Costa and McCrae, 1999) [117], a reduced version of the well-known NEO-PI-R. The questionnaire consists of 60 items to assess personality according to the “big five” model (Neuroticism, Extraversion, Openness, Friendliness and Responsibility).

### 3.6. Experimental Design for Mood Prediction

This study was carried out thanks to the dataset collected in [10]. Moreover, the individual interviews performed with each candidate were used to collect demographic information and the following questionaries were used to assess their mental health: STAI [109], Hamilton anxiety scale [110], Yesavage geriatric depression scale [111], MEC-35 [112], Ryff scale of psychological well-being [115], global deterioration scale [113], Katz index [114], PANAS [116]. After the interview, the EMA App was installed in their personal smartphone, and they received an Empatica E4 wristband. Data collection was collected for 15 consecutive days. During this time, the participants were asked to wear the device during the day and remove it before sleeping. They also received instructions on how to answer the daily EMAs sent to their smartphones. After the data collection, they came back to the laboratory and repeated the personal interview. They were asked to complete a satisfaction survey and to return the E4 wristband. According to the questionnaires, they did not show any symptoms of anxiety, mental disorder, or disability nor were there any significant differences with respect to the first questionnaire, maintaining stable results. Demographical information and the collected data are summarized in Table 12.

## 4. Results

### 4.1. Mood Prediction Resutls

We built a mood classifier using the support vector machine SVM (Support Vector Machines) library libSVM [118] using a radial basis function (RBF). Parameters C and γ were obtained by grid-search using cross-validation. The feature vectors were scaled in the range [−1, 1].

We propose two experiments to assess the ability of our model to solve the proposed problem of emotion recognition. Datasets were balanced due to lack of labels for certain mood states. It was discarded all samples which represent less than 10% of the total dataset. In addition, arousal and happiness dimension values were classified for the original sampled datasets, that is, without dropping non-representative mood states. The first experiment consists of splitting the dataset in two subsets, so that 75% of the samples are used for training and the remaining 25% are used for testing (Figure 11). Additionally, we include confusion matrices as a qualitative result (Figure 12, Figure 13, Figure 14 and Figure 15).

The second experiment is named as leave-one(day)-out. It consists of using one full day of data to test the model and the rest of days for training the classifier. The accuracy was computed as the mean of all tested days (Figure 16). The leave-one(day)-out approach shows an exciting interest in assessing the ability of the classifier to predict tomorrow’s emotional well-being based on an historic data register.

### 4.2. Robotic Platform Continuous Operation Results

This section provides the results obtained on the charging and discharging patterns of the robotic platform battery and the continuous operation experiments as well as the main observations made during the process. The testing environment was a real house. It was decided to perform the mapping in two different parts of the house to test the mapping task (Figure 17).

The scenario shown in Figure 17b was selected for the battery charging and discharging experiments and the continuous operation experiments with navigation according to an hourly routine. To prove that the script developed to monitor the battery status worked correctly, four experiments were carried out to see if the battery discharge profile resembled the one provided by the manufacturer (Figure 18).

Figure 19 shows the evolution of the battery status on a normal day without failures at the time of the auto docking task. Figure 20a shows the first continuous operation experiment and Figure 20b the second one.

## 5. Discussion

### 5.1. Mood Prediction

The first experiment consisted of dividing the dataset in 75%–25% proportions for training and testing, respectively. We repeated this process ten times to ensure a certain degree of confidence. Figure 11 shows the accuracy obtained from training a classifier to predict mood mental states. We tried different EMA window of values 30, 60 and 120 min. The highest accuracies achieved in this first experiment correspond to the metrics obtained by patient three (P3) in estimating mood in 30-, 60-, 120-min windows: 83.44% ± 1.6; 86.68% ± 1.43; 88.75% ± 0.78, respectively. It is interesting to note that patient three (P3) had the highest EMA-answering average, as reported in Table 12.

When inspecting a confusion matrix, one is seeking the main diagonal to obtain the highest possible values. This would mean great precision of the trained algorithm. It can be observed in Figure 12, Figure 13, Figure 14 and Figure 15, that the highest values are under the main diagonal. Nevertheless, there is also a certain degree of confusion among the different classes. Figure 12 shows confusion between contentment and pleasure in patient P1, between sleepiness and pleasure. Figure 13 shows that the algorithm is misclassifying pleasure or arousal with excitement in patient P2 and contentment is confused with pleasure or arousal. Figure 14 shows for patient P3 a confusion between arousal with excitement, and between depression and excitement. Finally, Figure 15 confuses pleasure with excitement or sleepiness in patient P2. By inspecting the circumplex model introduced by Russel [76], all the confused aforementioned emotions are placed in the circle quite close to each other. As most of the mood states contain a high pleasure value (representing positive moods), we can conclude that to achieve a fine-grained algorithm able to fit data close to each other in the feature space we would need to collect more samples, which would increase the complexity of the algorithm. The case of patient P3, who confused depression with excitement, also caught our attention. These two mood states are far from each other in Russel’s circumplex model [76]. Future research will inspect the nature of the depressed- and excited-labelled raw vectors and how they spatially fall into the dimensionally reduced feature space.

To validate the precision of our trained algorithm for this second experiment, we compared it with the study by R. Likamwa et al. [119], which shares some similarities with ours. To carry out this comparison, we learnt a model for the pleasure value, which is the dimension used in [119]. We are conscious that to compare two learning algorithms one should use the same dataset. However, the approach followed by [119] is exactly what we are pursuing in this second experiment. This comparison is used to confirm that the obtained results follow the correct direction. Moreover, the work in [119] uses periods of 10 and 20 days (about 3 weeks) to train their models. We used the available number of days for each participant, which as it can be observed it differs from those reported by [119]. Figure 16 shows our results laying within the reported results by [119]. This led us to conclude that, with a bigger dataset, that is, collecting more signals throughout more days, higher accuracy metrics will be achieved.

### 5.2. Robotic Platform

The adaptation of Hokuyo UST-10LX for mapping and navigation of the robotic platform is correct, as can be seen in Figure 17. The areas not so well defined on the map are inaccessible to the robot, for example, the toilets in Figure 17. In the case of (b) there is no defined area. This is because there is a step at the entrance to the bathroom that the robot could not go beyond and thus could not map that area. as in the case in (a). In general, good results were obtained in terms of mapping although the main disadvantage was that it had to be teleoperated.

The shape is similar to that provided by the manufacturer as can be seen in the discharge profiles in Figure 18. The main difference is that the manufacturer’s graphs show 9 h of operation for a single 4S2P battery while our two 4S2P batteries operated in a range between 6 and 7 h. This result is not so strange if you consider that the manufacturer’s curves have only the base consumption, while we in addition had a computer and Hokuyo UST-10LX to explain the reduced autonomy.

The main difference between the curves in Figure 18a,b is that in the case of (a), the battery was charged while the computer was on. This continuous consumption prevented it from reaching full charge. In the case of (b), the base was completely switched off for charging so that the start of the graphs in B is higher than in A and in addition the discharge lasts longer.

The autonomous navigation tests using the Hokuyo UST-10LX were thus successful. Using the ROS navigation stack, it was possible to navigate autonomously throughout the house, avoiding both dynamic and static obstacles. The main navigation problem was at the doors and there were times when the robot got stuck in doorways because its navigation algorithm determined there was going to be an imminent collision.

On the other hand, the experiment with continuous operation was not very encouraging. As can be seen in Figure 20, the main problem occurred in executing the autodocking task. The specification for the most successful autodocking is to have a clean area 2 meters wide by 5 meters long around it. It should be noted that initially, an attempt was made to reduce this space to see how reliable the algorithm was by reducing this specification and it should be remembered that not every house has a clear space of 2 × 5 meters. The results obtained are shown in Figure 20a. As can be seen, compared to Figure 20a, there were many problems with autodocking. To remedy this, the station was relocated to a larger area that met the specifications. From the results shown in Figure 20b, it can be seen that the autodocking algorithm performed reasonably better. However, it was still flawed and, at times, required human intervention.

## 6. Conclusions

Some conclusions can be drawn in the order of the descriptions of the systems. Starting with a smart home, the system was in operation for 21 days without failures with the devices connected continuously. It should be noted that there were periods in which the system was without power due to power cuts and that the devices reconnected normally after restarting Raspberry Pi. Based on this experience, it can therefore be concluded that the system works satisfactorily and offers robustness connectivity.

It was also concluded that the robotic navigation and autodocking algorithms used are not sufficiently robust. Although the navigation algorithm works well in most cases, the robot needs human intervention when it fails. Some of the scenarios in which failures occurred were when navigating through narrow doorways and corridors. The autodocking algorithm runs normally if the surrounding conditions required by the algorithm to function correctly are met (2 × 5 empty area in front) and when it fails manual intervention is required. These results were observed during the evolution of the continuous operation experiments. In the discharge profiles shown in Figure 20, it can only be detected that the system has failed due to an abnormal discharge profile. As discussed, the most common faults causing this situation have been navigation in narrow areas and failures in performing autodocking. To address navigation problems, we are currently working on the implementation of semantic navigation algorithms to improve autonomous navigation in a changing environment and on the use of computer vision to detect the docking station.

The model shows a promising ability to predict mood values by means of only physiological data for the machine learning application for mood prediction. Experiments show that a 120-min EMA window works significantly better in predicting mood values, while the accuracy in predicting arousal, pleasure and mood values are comparable to, or better than, other methods reported in the literature.

The performance metrics obtained for the experiment leave-one(day)-out exhibits a path for future work, although its accuracy is similar to that found in other works. Reference [119] reported that, to achieve a performance of over 80%, 40 to 60 days were needed (about 2 months) so that our datasets need to be extended. The present system is endowed only with physiological data. Future research can combine different data modalities such as smartphone information (agenda events, screen use, etc.) or audio signals to provide greater model robustness.

As the learnt mood models also need generalization and transferability it is proposed to increase the complexity of the model’s classifier by using Deep Learning techniques as more data are collected. 

The experimental data in this work were collected during daily activities with no restrictions on the participants’ behavior, which shows that our technology is viable for daily life use. The results obtained seem to indicate that the recognition of the user’s mood is sufficiently effective to be able to trigger the coaching actions that the team of psychologists considers appropriate for each user in each situation.

In terms of applying these technologies, research in artificial intelligence and home automation is especially relevant for therapeutic applications in mental health services. The interventions were especially directed towards caring for the elderly. It should be noted that our project focused on innovating by providing a responsible approach (under the supervision of a trained mental health professional) and taking into account the ethical implications of incorporating artificial intelligence and home automation in people’s lives (ecological sensors in the house so as not to interfere, the use of bracelets etc.). The most important thing is that our project escapes from the laboratory since it allows the subjects to carry out their daily activities by means of smart devices. Another of the fundamental aspects is the daily monitoring, after a previous evaluation, to be able to evaluate moods and mental states and to be able to make diagnoses where appropriate. From an ethical and responsible perspective, this project will bring important benefits from the application of robotics and artificial intelligence to mental health, which will allow new modes of treatment, opportunities to involve hard-to-reach populations, improve adherence to patient response and free up time for specialists through combined care models. That is why we argue that the union of artificial intelligence and home automation is a promising approach in the entire field of mental health, especially in innovative mental health care.

## Figures and Tables

**Figure 1 sensors-21-06865-f001:**
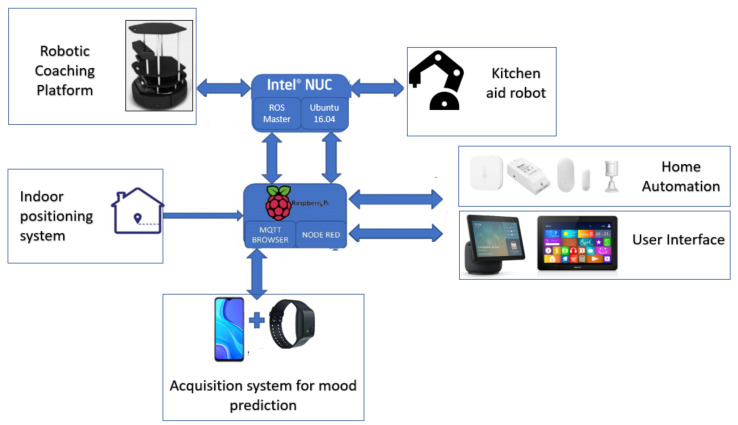
Hardware schematic of the system (only some representative elements of each subsystem are shown).

**Figure 2 sensors-21-06865-f002:**
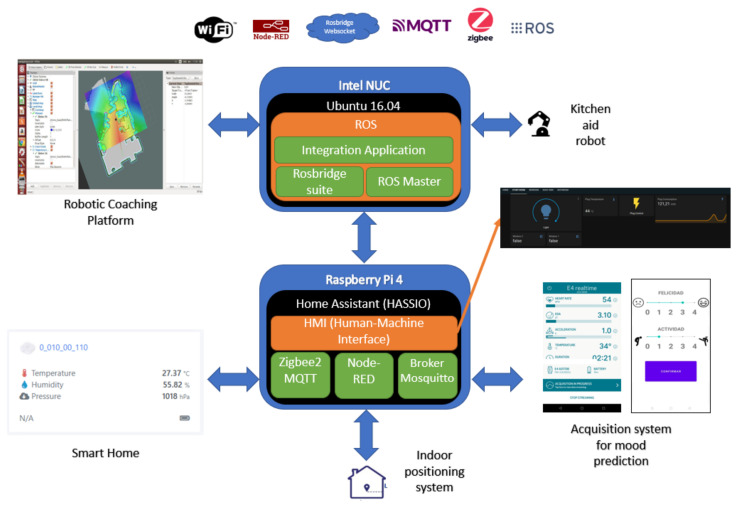
Software schematic of the system.

**Figure 3 sensors-21-06865-f003:**
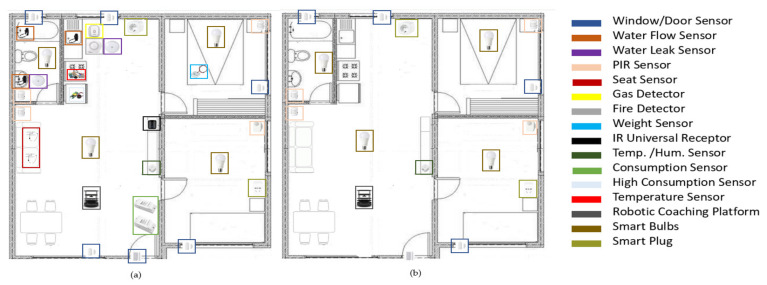
(**a**) Map of proposed elements in the final system. (**b**) Map of test system elements.

**Figure 4 sensors-21-06865-f004:**
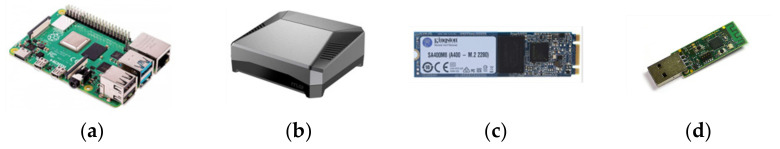
(**a**) Raspberry Pi, (**b**) Raspberry Pi Aluminum Case, (**c**) SSD Memory Disk, (**d**) USB Dongle cc2531.

**Figure 5 sensors-21-06865-f005:**
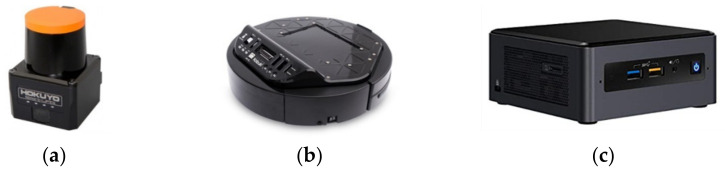
(**a**) Hokuyo UST-10LX; (**b**) IClebo robotic base; (**c**) INTEL NUC computer.

**Figure 6 sensors-21-06865-f006:**
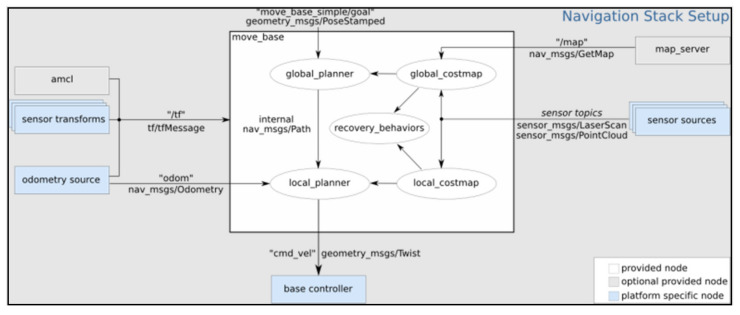
ROS navigation stack setup [95].

**Figure 7 sensors-21-06865-f007:**
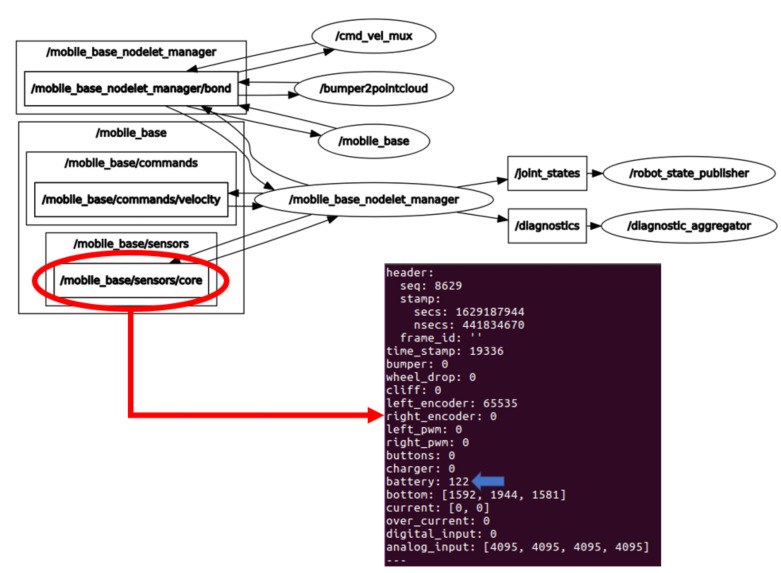
ROS nodes and topics from turtlebot_bringup.

**Figure 8 sensors-21-06865-f008:**
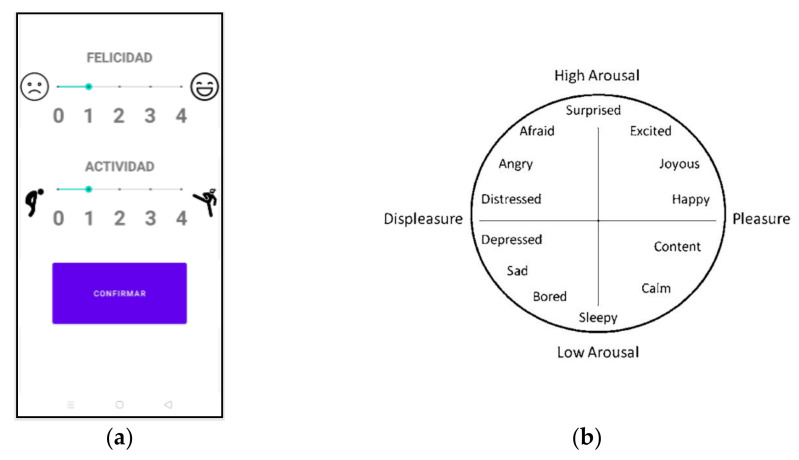
(**a**) Mobile application to collect mood states labels (on the left); (**b**) Affection state model [76].

**Figure 9 sensors-21-06865-f009:**
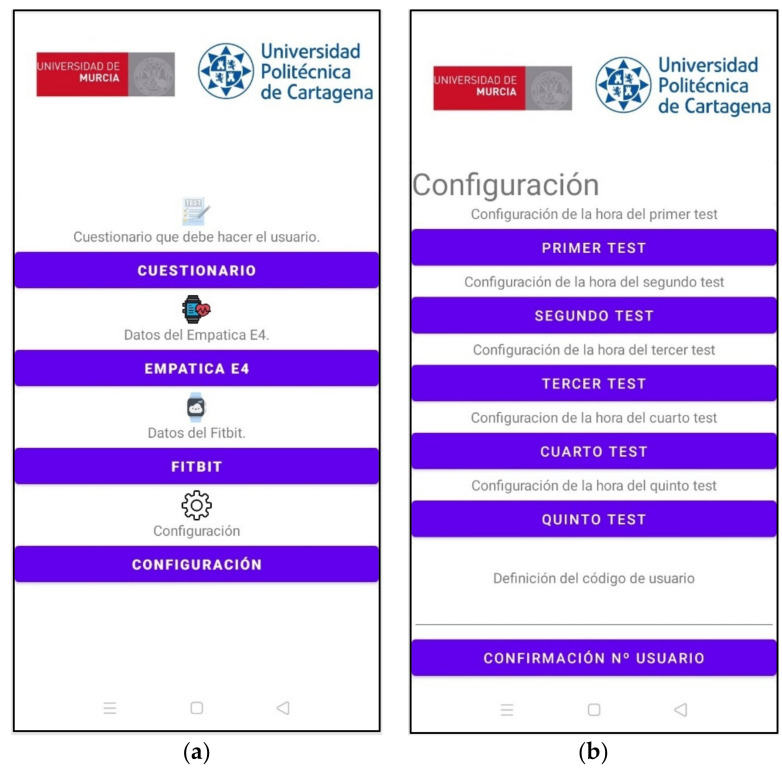
(**a**) Administrator mode main menu. (**b**) Administrator mode test time settings.

**Figure 10 sensors-21-06865-f010:**
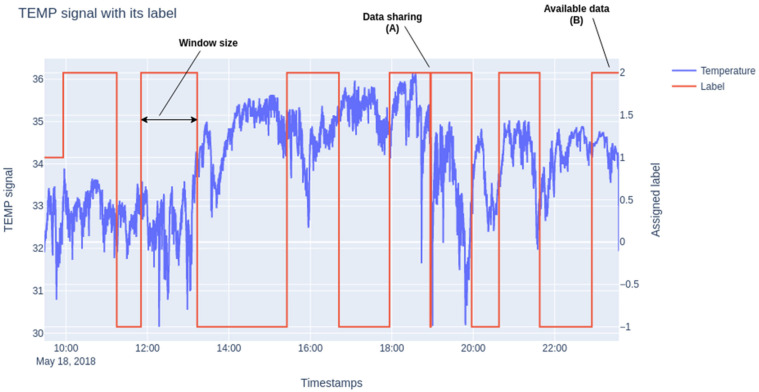
Illustration of the extrapolation of the mood states obtained from the EMA answers received in one day with a 60-min centered window for temperature signal. Red line represents the mood state assigned (−1 represents no mood state). Situation A is when two consecutive EMA answers are given within an interval of time less than half the size of the EMA window. Situation B is when there is no available data in the initial or last EMA answered.

**Figure 11 sensors-21-06865-f011:**
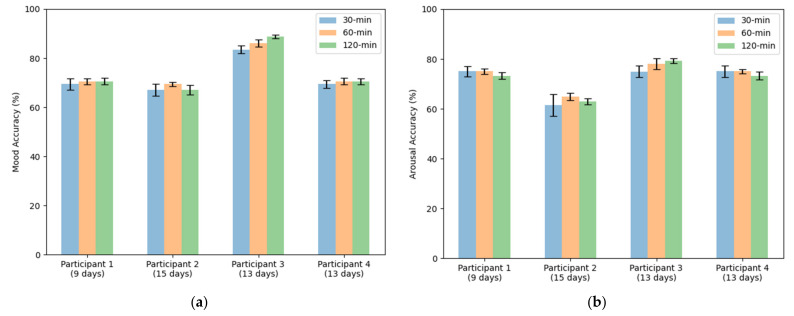
Accuracy metric obtained from predicting mood (**a**), arousal (or activeness) (**b**), and pleasure (or happiness) (**c**) values. Error bars represent standard deviation.

**Figure 12 sensors-21-06865-f012:**
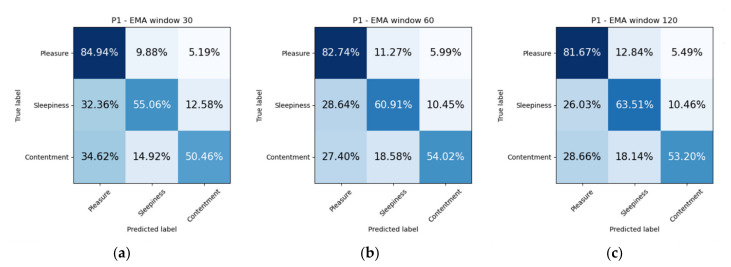
Confusion matrix for patient P1 for windows (**a**) 30-, (**b**) 60-, (**c**) 120-min. Predicted mood states: Pleasure, Sleepiness and Contentment.

**Figure 13 sensors-21-06865-f013:**
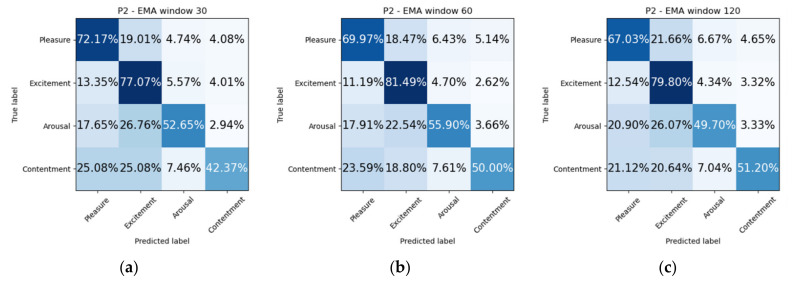
Confusion matrix for patient P2 for windows (**a**) 30-, (**b**) 60-, (**c**) 120-min. Predicted mood states: Pleasure, Excitement, Arousal and Contentment.

**Figure 14 sensors-21-06865-f014:**
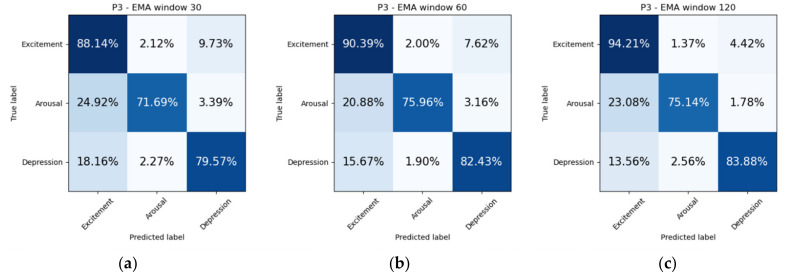
Confusion matrix for patient P3 for windows (**a**) 30-, (**b**) 60-, (**c**) 120-min. Predicted mood states: Excitement, Arousal and Depression.

**Figure 15 sensors-21-06865-f015:**
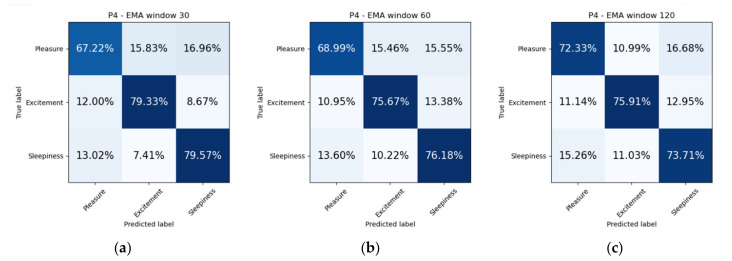
Confusion matrix for patient P4 for windows (**a**) 30-, (**b**) 60-, (**c**) 120-min. Predicted mood states: Pleasure, Excitement and Sleepiness.

**Figure 16 sensors-21-06865-f016:**
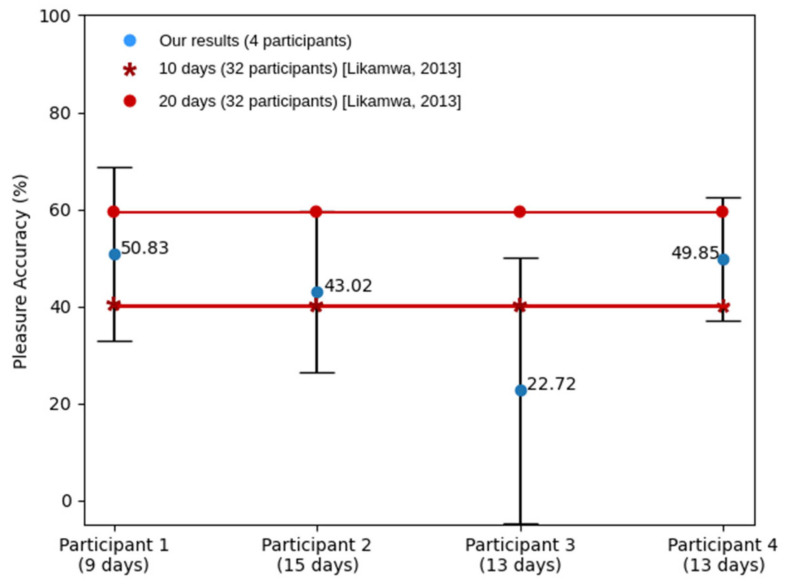
Accuracy metric obtained for the leave-one(day)-out experiment to predict pleasure (or happiness) values. Error bars represent standard deviation. Red asterisk and Circle dots represent the mean pleasure accuracy for datasets of 10 and 20 days respectively stated by R. Likamwa et al. [119].

**Figure 17 sensors-21-06865-f017:**
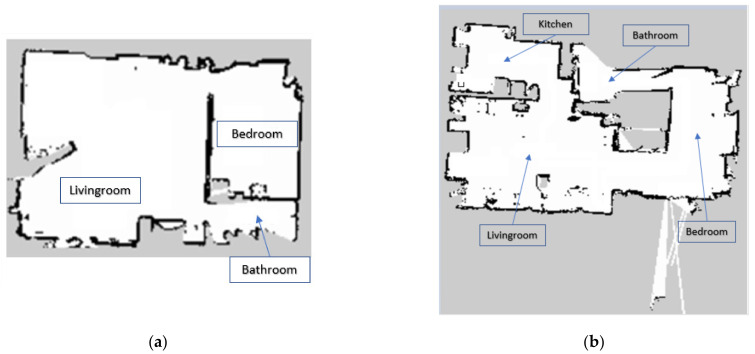
(**a**) Map of the first zone of the house. (**b**) Map of the second zone of the house.

**Figure 18 sensors-21-06865-f018:**
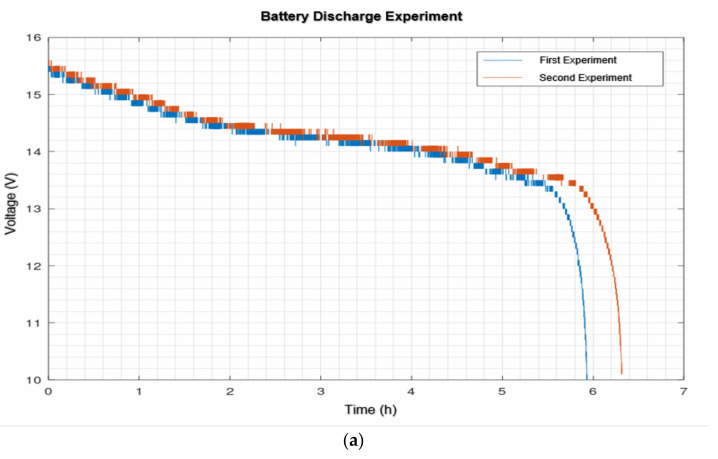
Battery discharge profiles: (**a**) First and second experiments, (**b**) Third and fourth experiments.

**Figure 19 sensors-21-06865-f019:**
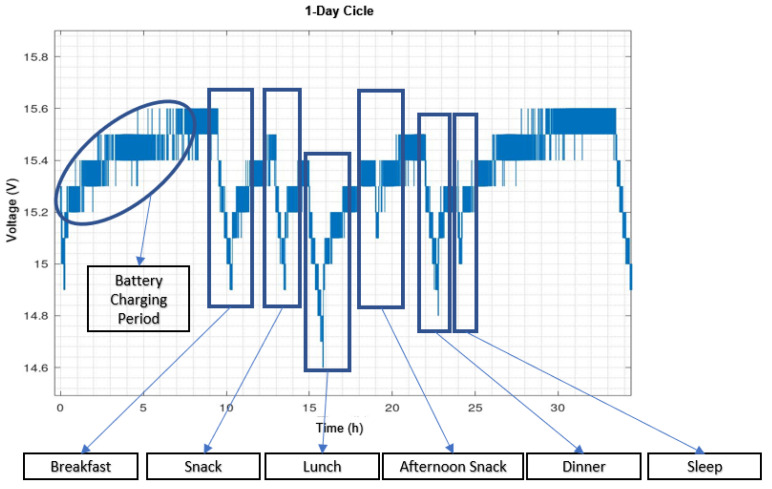
Battery charge and discharge profile for a normal operating day.

**Figure 20 sensors-21-06865-f020:**
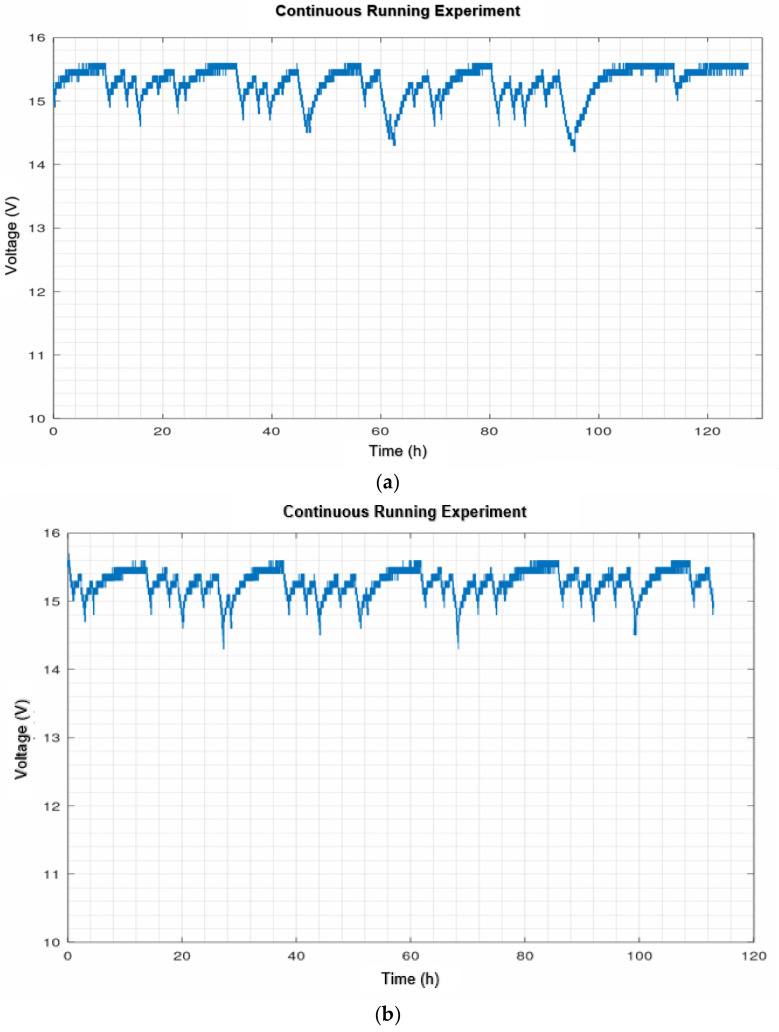
(**a**) First continuous operation experiment. (**b**) Second continuous operation experiment.

**Table 1 sensors-21-06865-t001:** General description of the system elements.

General	Elements	Description
Sensors	Smart home	Commercially available and self-built sensors for monitoring the user’s daily activity.
	Empatica E4	A medical device, such as an activity bracelet, for the acquisition of physiological signal data.
	Robotic Platform Sensors	Hokuyo UST-10LX LIDAR sensor for robot mapping and navigation.
		Set of proprioceptive sensors installed in the iClebo Kobuki base (bumpers, gyroscope...).
Control	Raspberry Pi	Computational node that executes the logic of the home automation section. It also serves as an integration node.
	Intel NUC	Computational node that executes the logic of the integration software application. Artificial intelligence algorithms for coaching strategies.
	Intel NUC	Robot controller
Actuator	Robotic Platform	A robotic system that will navigate autonomously through the house and interact with the user by providing them with coaching strategies.
	Smart home	Elements such as shutter motors, lights and smart plugs.

**Table 2 sensors-21-06865-t002:** Proposed monitoring activities and devices.

Activity	Devices
Sleeping hours, physical activity, frequency of toilet visit	Activity wristband
Social interaction monitoring	Telephone call screening device (landline) or call logging services (mobile)
Drug consumption	Electronic pill dispenser
Proper nutrition	Fridge monitoring
Personal hygiene	Water flow detectors
Movement between rooms	Motion detectors and positioning system
Occupancy of armchairs/chairs	Pressure detectors
Bed occupancy	Load cells
Proper condition of the house	Temperature/Humidity sensor
TV viewing	Smart TV service integration
Opening of drawers, doors, windows...	Switch and windows/doors sensor
Use of household appliances	Power consumption sensors

**Table 3 sensors-21-06865-t003:** Home automation system devices.

Sensor Name	Model	Protocol	Description	Inc.
Mi Motion Sensor	YTC4041GL	Zigbee	PIR sensor from Xiaomi	Y
IKEA PIR Sensor		Zigbee	PIR sensor from IKEA	Y
Temp/Hum Aqara	WSDCGQ11LM	Zigbee	Temperature/Humidity sensor from Aqara	Y
Mi Window/Door Sensor	YTC4039GL	Zigbee	Window/Door sensor from Xiaomi	Y
LIDL Window/Door sensor	TY0203	Zigbee	Window/Door sensor from LIDL	Y
Heiman Gas Detector	HS1CG	Zigbee	Natural gas detector from Heiman	N
Aqara Leak Water Detector	SJCGQ11LM	Zigbee	Leak water detector from Aqara	N
Xiaomi Smoke Detector	JTYJ-GD-03MI/BB	Zigbee	Fire and smoke detector from Xiaomi	N
Smart Plug Xiaomi	ZNCZ04CM	Zigbee	Smart Plug from Xiaomi	Y
Smart Plug LIDL	HG06337	Zigbee	Smart plug from LIDL	Y
Smart Bulb	LED1836G9	Zigbee	Smart Bulb E27 806 lumens from IKEA.	Y
Sonoff POW	IM171130001	WiFi	Consume monitoring fom Sonoff	N
Broadlink RM-Mini3	RM-MINI3	IR	Universal IR gateway	N
Robotic Platform Sensor	TURTLEBOT II	WiFi	Robotic base sensor, LIDAR Hokuyo UST-10LX and Orbecc Astra	Y
Flow Sensor	Own development	WiFi	Flow water sensor for kitchen and bathroom use.	N
Seat Sensor	Own development	WiFi	Sensor for monitoring the user’s time spent in a seated position	N
Bed Weight Sensor	Own development	WiFi	Sensor for monitoring user’s weight and time in bed	N
Large Consumption Sensor	Own development	WiFi	Sensor for high power consumption devices in the home. The PZEM-004T shall be used.	N
Thermocouple vitro/stove	Own development	WiFi	Sensor for monitoring the use of the cooker or hobs	N
Telephone Sensor	Own development	WiFi	Sensor to monitor social interaction via landline phone	N
Indoor Positioning System	Own development	WiFi	Indoor location system with decaWave	N

**Table 4 sensors-21-06865-t004:** Navigation routines.

Breakfast	Bedroom	08:00
	Bathroom	08:05
	Livingroom	08:15
	Kitchen	08:17
	Bathroom	08:37
	Livingroom	08:45
	Docking Station	08:47
Snack	Livingroom	11:30
	Kitchen	11:33
	Bathroom	11:47
	Livingroom	12:00
	Docking Station	12:02
Lunch	Livingroom	13:30
	Kitchen	13:33
	Bathroom	14:00
	Livingroom	14:10
	Docking Station	14:12
Afternoon Snack	Livingroom	17:30
	Kitchen	17:32
	Bathroom	17:47
	Livingroom	17:57
	Docking Station	18:00
Dinner	Livingroom	20:30
	Kitchen	20:33
	Bathroom	21:02
	Livingroom	21:10
	Docking Station	21:12
Sleep	Livingroom	22:30
	Bedroom	22:32
	Bathroom	22:35
	Bedroom	22:45
	Docking Station	22:47

**Table 5 sensors-21-06865-t005:** Empatica E4 built-in sensors.

Signal	E4 Built-In Sensor	Range	Resolution Sample Rate
Accelerometer	3-axis accelerometer	−2 g +2 g	8 bit 32 Hz
Temperature	Infrared thermopile	−40 ° +115 °C	14 bit 0.02 °C 4 Hz
Electrodermal Activity	AC Current source 100 uA @ 8 Hz/100 uSiemens	0.01 µSiemens 100 µSiemens	14 bit 900 pSiemens 4 Hz
Heart Rate	Photoplethysmography Blood Volume Pulse 4 LEDs (2 × Green, 2 × Red) 2 PhotodiodesTotal 15.5 mm^2^ sensitve area.		0.9 nW 64 Hz

**Table 6 sensors-21-06865-t006:** Signal preprocessing.

Signal	Filtering
Accelerometer	Band-pass filter (0.2 Hz–10 Hz) 3rd order Butterworth
Temperature	Threshold @ 2 °C
Electrodermal Activity	Low-pass filter (0–1.5 Hz) 3rd order Butterworth EDA, SCLR and SCR Extraction [99]
Heart Rate	HR resampled at 1 Hz Normalized to first 10 min/day [10]

**Table 7 sensors-21-06865-t007:** Accelerometer features.

Domain	Measurement	Features		Reference
Time	Maximum value of the rectified segment	MAX	4	[100]
	90th percentile of the rectified segment	P90	4	
	Variance of the segment	VAR	4	
	Mean Absolute Deviation of the segment	MAD	4	
	Norm of the segment	Norm	4	
	Different between maximum amplitude and mean of signal segment	AMP	4	[101]
	Minimum value of signal segment	MIN	4	
	Standard deviation of signal segment	STD	4	
	Root Mean Square of signal segment	RMS	4	
	Means of the Absolute Values of the First Differences of the raw signal segment	MAVFD	4	[85]
	Means of the Absolute Values of the First Differences of the normalized signal segment	MAVFDN	4	
	Means of the Absolute Values of the Second Differences of the raw signal segment	MAVSD	4	
	Means of the Absolute Values of the Second Differences of the normalized signal segment	MAVSDN	4	
Frequency	Mean value of Power Spectral Density	FM	4	[102]
	Standard deviation of Power Spectral Density	FSTD	4	
	25-percentile of Power Spectral Density	FP25	4	[10]
	50-percentile of Power Spectral Density	FP50	4	
	75-percentile of Power Spectral Density	FP75	4	

**Table 8 sensors-21-06865-t008:** Skin Temperature Features.

Domain	Measurement	Features		Reference
Time	Mean temperature of the segment	MT	1	[103]
	Slope of a fitted regression line of the segment	SRL	1	
	Intercept of a fitted regression line of the segment	IRL	1	
	Standard deviation of signal segment	STD	1	[101]
	Means of the Absolute Values of the First Differences of the raw signal segment	MAVFD	1	[85]
	Means of the Absolute Values of the First Differences of the normalized signal segment	MAVFDN	1	
	Means of the Absolute Values of the Second Differences of the raw signal segment	MAVSD	1	
	Means of the Absolute Values of the Second Differences of the normalized signal segment	MAVSDN	1	
Frequency	Mean value of Power Spectral Density (PSD)	FM	1	[102]
	Standard deviation of Power Spectral Density (PSD)	FSTD	1	
	25-percentile of Power Spectral Density (PSD)	FP25	1	[10]
	50-percentile of Power Spectral Density (PSD)	FP50	1	
	75-percentile of Power Spectral Density (PSD)	FP75	1	

**Table 9 sensors-21-06865-t009:** Heart rate features.

Domain	Measurement	Features		Reference
Time	Maximum value of the rectified segment	MAXHR	1	[100]
	90th percentile of the rectified segment	P90HR	1	
	Variance of the segment	VARHR	1	
	Mean Absolute Deviation of the segment	MADHR	1	
	Norm of the segment	normHR	1	
	Mean value of signal segment	Mean	1	[101]
	Minimum value of signal segment	MIN	1	
	Standard deviation of signal segment	STD	1	
	Means of the Absolute Values of the First Differences of the raw signal segment	MAVFD	1	[85]
	Means of the Absolute Values of the First Differences of the normalized signal segment	MAVFDN	1	
	Means of the Absolute Values of the Second Differences of the raw signal segment	MAVSD	1	
	Means of the Absolute Values of the Second Differences of the normalized signal segment	MAVSDN	1	
	Mean of the Smooth HR	SM	1	
	Mean of the First Differences	MFD	1	
Frequency	Very Low Absolute Spectral Power	aVLF	1	[104]
	Low Absolute Spectral Power	aLF	1	
	High Absolute Spectral Power	aHF	1	
	Total Absolute Spectral Power	aTotal	1	
	Very Low Frequency Spectral Power in Percentage	pVLF	1	
	Low Frequency Spectral Power in Percentage	pLF	1	
	High Frequency Spectral Power in Percentage	pHF	1	
	Low Frequency Spectral Power Normalize to Total Power	nLF	1	
	High Frequency Spectral Power Normalize to Total Power	nHF	1	
	Ratio of Low Frequency to High Frequency	LFHF	1	
	Very Low Frequency Peak	peakVLF	1	
	Low Frequency Peak	peakLF	1	
	High Frequency Peak	peakHF	1	

**Table 10 sensors-21-06865-t010:** Electrodermal activity features.

Domain	Measurement	Features		Reference
Time	Mean	MSC	3	[99]
	Standard deviation	SDSC	3	
	Maximum	MASC	3	
	Minimum	MISC	3	
	Dynamic range	DRSC	3	
	Mean of the first derivative	FMSC	3	
	Standard deviation of the first derivative	FDSC	3	
	Mean of the second derivative	SMSC	3	
	Standard deviation of the second derivative	SDSC	3	
	Arc length	ALSC	3	
	Integral	INSC	3	
	Normalized average power	APSC	3	
	Normalized root mean square	RMSC	3	
	Area-perimeter	ILSC	3	
	Energy-perimeter	ELSC	3	
	High order skewness	SKSC	3	
	High order kurtosis	KUSC	3	
	Central moment	MOSC	3	
	Means of the Absolute Values of the First Differences of the raw signal segment	MAVFD	1	[85]
	Means of the Absolute Values of the First Differences of the normalized signal segment	MAVFDN	1	
	Means of the Absolute Values of the Second Differences of the raw signal segment	MAVSD	1	
	Means of the Absolute Values of the Second Differences of the normalized signal segment	MAVSDN	1	
Frequency	Spectral power in bandwidths 0.1 to 0.2	F1SC	3	[99]
	Spectral power in bandwidths 0.2 to 0.3	F2SC	3	
	Spectral power in bandwidths 0.3 to 0.4	F3SC	3	
	Mean value of Power Spectral	FMSCR	3	[102]
	Standard deviation of Power Spectral Density	FSTDSCR	3	
	25-percentile of Power Spectral Density	FP25	3	[10]
	50-percentile of Power Spectral Density	FP50	3	
	75-percentile of Power Spectral Density	FP75	3	

**Table 11 sensors-21-06865-t011:** Information collected by the application.

Information	Description
Happiness level	Representation of the happiness level felt by the user at the time of the test on a discrete scale divided into 5 levels (0–4).
Activity level	Representation of the activity level felt by the user at the time of the test on a discrete scale divided into 5 levels (0–4).
Test hours	User-defined time register of time tests.
Reaction time	Timestamp at which the application notifies that the test is to be performed and timestamp at which the user accesses the notification.

**Table 12 sensors-21-06865-t012:** Participants’ details.

	P1	P2	P3	P4
Age	67	55	60	63
Gender	Male	Female	Male	Female
Days	9	15	14	14
Non-valid days	0	0	2	1
Total EMAs (Daily avg.)	42 (4.67 ± 0.67)	57 (3.8 ± 0.75)	64 (5.33 ± 1.84)	46 (3.54 ± 0.93)

## Data Availability

The data presented in this study are available on request from the corresponding author. The data are not publicly available due to privacy issues.

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
