# Peer review of "Robotic-Based Well-Being Monitoring and Coaching System for the Elderly in Their Daily Activities"

_sensors, 2021, doi:10.3390/s21206865_

Round 1

Reviewer 1 Report

The provided manuscript is prepared in a quite good manner, but it is too extensive. It includes quite a lot of aspects and due to this is hard to follow the main goal.  From my point of view, some major improving required. First of all, the title of the article looks like a review, and in the begging, it seems that it is a review article, but later it became clear that it is not a review. I strongly recommend clearly separate the literature review from the authors' contribution. Secondly, I recommend extending the description of the manuscript aim and provide more information about Robwell project. Thirdly, I recommend minimizing the amount of technical data, especially characteristics of well known and freely available components such as the Raspberry Pi controller. Fourthly, I recommend providing a clear comparison between proposed and existing systems in graphical or table form. Moreover, the authors provided a lot of experimental results, but conclusions are not supported by them, this must be corrected.

Minor inaccuracies:

An editing error in Line 77 and 78, they provide analogical statements with different references.

Figures 1 and 2 are too difficult. Fonts are too small and hardly readable. The font in figures 3,7,9, 18-20 also must be increased.

Author Response

First of all, I would like to thank you for your comments. Following your response criteria, comments will be answered by points:

1.- First of all, the title of the article looks like a review, and in the begging, it seems that it is a review article, but later it became clear that it is not a review. I strongly recommend clearly separate the literature review from the authors' contribution.

  • We realised that the title could be misleading, and it has been changed. On the other hand, it has been clarified in the abstract that the article is about the Robwell project. In this way, it is clearer that it is not a review.

2.- Secondly, I recommend extending the description of the manuscript aim and provide more information about Robwell project.

  • A description of the main points addressed in this article has been added at the end of the introduction. In addition, the Robwell project of the 66-74 and 105-128 line is also discussed.

3.- Thirdly, I recommend minimizing the amount of technical data, especially characteristics of well-known and freely available components such as the Raspberry Pi controller.

  • It has been decided to remove the tables containing information about the Raspberry, the Hokuyo laser sensor, the Intel NUC computer and the Kobuki robotic base. In addition, the coding followed for the elements of the home automation system has been removed.

4.- Fourthly, I recommend providing a clear comparison between proposed and existing systems in graphical or table form.

  • In order to make the comparison and the contributions of the proposed system compared to those mentioned in the state of the art, it has been decided to add the section "2.4 Contributions". It has not been found necessary to use a graphical representation.

5.- Moreover, the authors provided a lot of experimental results, but conclusions are not supported by them, this must be corrected.

  • Initially, section 6 was oriented as the ideas derived from the discussion of the data. However, references to the data have been added to support these ideas. In the case of home automation, no data has been provided as the system has been tested experimentally in a real house and no performance statistics have been obtained. However, the user experience of the system is available. In the case of robotics, the conclusions are based on the charging and discharging profiles of the batteries during continuous operation tests. In the case of the conclusions of the artificial intelligence part, they have been justified with the confusion matrices presented in the results section.

6.-  An editing error in Line 77 and 78, they provide analogical statements with different references.

  • That was an editing error. Fixed.

7.-  Figures 1 and 2 are too difficult

  • The system presented includes a multitude of elements. The aim of these diagrams was to collect as much information as possible. However, they have been simplified and enlarged.

8.- The font in figures 3,7,9, 18-20 also must be increased.

  • The font of all figures has been enlarged.

Reviewer 2 Report

The paper proposes a system for healthcare monitoring of elders which uses machine learning algorithms for predicting the mood of an elder based on physiological signals. In particular, the system integrates sensors and actuators for monitoring a person's daily activities, habits and mood on one hand and a robotic platform for enforcing emotional coaching strategies.

The paper addresses a topic of great interest and proposes an interesting solution that could be very helpful for the elderly especially in the current times affected by pandemics.

The paper is well organized, however I consider that the authors should make the following improvements to make it suitable for being published:

For example, in the related work section authors present several approaches similar to the one presented in the paper, however a comparison with these approaches is missing in order to highlight the contributions
of the paper. Therefore I suggest authors to clearly emphasize their contributions compared to the current state of the art.

I also consider that the presentation of the technique for predicting the mood of a person is not very clear - for example section 3.4.3 is entitled "Signal classification" but the section just describes what the authors used for predicting the mood and not how they did it. Also, authors should give more details about the features they used and how they aggregated data coming from heterogeneous sources such as questioneers and wristband.

The authors should also adjust the tables because they seem to be too large and out of the paper's margins (e.g. table 1 on page 8, table 3 on page 11, etc.), and should use the same font size in all tables. 

The paper also contains some minor language mistakes that the authors should correct (e.g. "If strange behavior or a low mood is detected, emotional coaching strategies are proposed by a small robot the size of a robot vacuum cleaner [9] coaching strategies are proposed by a small robot the size of a robot vacuum cleaner [4]" on page 2; "In this way, the sick, disa-bled" on page 3, etc.).

Finally, authors should refer all the figures in the text - for example Figure 4 is not referred  in the text.

Author Response

First of all, I would like to thank you for your comments. Following your response criteria, comments will be answered by points:

1.- For example, in the related work section authors present several approaches similar to the one presented in the paper, however a comparison with these approaches is missing in order to highlight the contributions of the paper. Therefore I suggest authors to clearly emphasize their contributions compared to the current state of the art.

  • In order to make the comparison and the contributions of the proposed system compared to those mentioned in the state of the art, it has been decided to add the section "2.4 Contributions".

2.- I also consider that the presentation of the technique for predicting the mood of a person is not very clear - for example section 3.4.3 is entitled "Signal classification" but the section just describes what the authors used for predicting the mood and not how they did it. Also, authors should give more details about the features they used and how they aggregated data coming from heterogeneous sources such as questioneers and wristband.

  • We agree that due to the structure, it can be confusing following the integration and aggregation of all model's parts, but we strictly followed the journal's structure to write the paper. It has been done modifications in section 3.4.3, highlighting and clarifying the aggregation of the different data sources and the purpose of it. We hope that now it easier to the reader to link between feature extraction from the wristband signals and the questionnaires done via the smartphone APP. Regarding the comment of giving more details about the features used, we invite the reviewer to check section 3.4.1.

3.- The authors should also adjust the tables because they seem to be too large and out of the paper's margins (e.g. table 1 on page 8, table 3 on page 11, etc.), and should use the same font size in all tables.

  • We agree with these observations. The tables have been revised, all put in the same font and adjusted to the margins.

4.- The paper also contains some minor language mistakes that the authors should correct (e.g. "If strange behavior or a low mood is detected, emotional coaching strategies are proposed by a small robot the size of a robot vacuum cleaner [9] coaching strategies are proposed by a small robot the size of a robot vacuum cleaner [4]" on page 2; "In this way, the sick, disa-bled" on page 3, etc.).

  • We agree with these observations. Most of these errors are due to editing. The article has been reviewed and corrected.

4.- Finally, authors should refer all the figures in the text - for example Figure 4 is not referred  in the text.

  • All figures and tables have been checked to ensure that they are referenced.

Round 2

Reviewer 1 Report

Authors reacted to all remarks, quality of the manuscript is improved. In my opinion now it suits for publication.